# Cashew Nutshells: A Promising Filler for 3D Printing Filaments

**DOI:** 10.3390/polym15224347

**Published:** 2023-11-07

**Authors:** María José Paternina Reyes, Jimy Unfried Silgado, Juan Felipe Santa Marín, Henry Alonso Colorado Lopera, Luis Armando Espitia Sanjuán

**Affiliations:** 1Engineering, Science and Technology Research Group, Mechanical Engineering Department, University of Córdoba, Cr. 6 No. 76-103, Montería 230002, Córdoba, Colombia; mpaterninareyes34@correo.unicordoba.edu.co (M.J.P.R.); jimyunfried@correo.unicordoba.edu.co (J.U.S.); 2Grupo de Investigación Materiales Avanzados y Energía—MATyER, Instituto Tecnológico Metropolitano, Medellín 050012, Antioquia, Colombia; jfsanta@unal.edu.co; 3CCComposites Laboratory, University of Antioquia, Calle 67 No. 53-108, Medellín 050010, Antioquia, Colombia; henry.colorado@udea.edu.co

**Keywords:** natural fiber-reinforced polymers, cashew nutshell particles, 3D printing filament

## Abstract

Cashew nutshells from the northern region of Colombia were prepared to assess their potential use as a filler in polymer matrix filaments for 3D printing. After drying and grinding processes, cashew nutshells were characterized using scanning electron microscopy (SEM), attenuated total reflectance Fourier-transform infrared (ATR-FTIR), and thermogravimetric analyses (TGA). Three different filaments were fabricated from polylactic acid pellets and cashew nutshell particles at 0.5, 1.0, and 2.0 weight percentages using a single-screw extruder. Subsequently, single-filament tensile tests were carried out on them. SEM images showed rough and porous particles composed of an arrangement of cellulose microfibrils embedded in a hemicellulose and lignin matrix, the typical microstructure reported for natural fibers. These characteristics observed in the particles are favorable for improving filler–matrix adhesion in polymer matrix composites. In addition, their low density of 0.337 g/cm^3^ makes them attractive for lightweight applications. ATR-FTIR spectra exhibited specific functional groups attributed to hemicellulose, cellulose, and lignin, as well as a possible transformation to crystalline cellulose during drying treatment. According to TGA analyses, the thermal stability of cashew nutshell particles is around 320 °C. The three polylactic acid–cashew nutshell particle filaments prepared in this work showed higher tensile strength and elongation at break when compared to polylactic acid filament. The characteristics displayed by these cashew nutshell particles make them a promising filler for 3D printing filaments.

## 1. Introduction

A remarkable augmentation in the use of natural fibers as reinforcement or dispersed phase in polymer matrix composites to replace pure polymers or traditional reinforcements has attracted a great deal of attention in the past years; however, these bio-composites exhibit large variability in their properties. The electrical, mechanical, and rheological properties of date palm waste-derived biochar–polypropylene matrix composites were investigated [1]. The authors reported that the increase in biochar content decreased tensile strength and ductility, but increased the modulus of elasticity, the electrical conductivity, and the storage moduli of composites. The effect of fiber content and fiber treatment on the mechanical behavior of banana fiber–polyethylene matrix composites was assessed [2]. In all cases, the 20 weight percentage fiber addition increased tensile strength, modulus of elasticity, and elongation at break. However, both 40 and 60 weight percentages, as a function of treatment, increased and decreased these properties, a fact attributed to the different chemical composition (lignin and cellulose contents) and the degree of crystallinity produced on the fibers by each treatment. Other researchers prepared bio-composites by compounding unmodified and acylated cork powder with polylactic acid (PLA), and polycaprolactone (PCL) matrices at 1, 5, 10, 20, and 30 weight percentages of powder [3]. The results indicated that unmodified and acylated cork contents equal or lower to 10 mass percentage slightly affected the modulus of elasticity of both composites, whereas a considerable decrease in this property was observed for 20 and 30 mass percentages. In contrast, tensile strength decreased with the augment on cork content, regardless of treatment and polymer matrix. The weathering performance of bio-composites has been evaluated by some researchers. The effect of moisture on the mechanical properties of PLA matrix composites reinforced with ramie, flax, and cotton fibers at 10, 20, 30, 40, and 50 mass percentages was reported [4]. Independently of moisture and fiber content, the fiber addition improved the flexural properties of composites; however, impact behavior decreased with increasing fiber content. Furthermore, the weathering resistance of bio-composites produced by the addition of microcrystalline cellulose and nutshell fibers into a high-density polyethylene matrix was investigated [5]. A weathering test was used to simulate outdoor degrading factors for 672 h. It was reported that bio-composites with nutshell fibers were less affected by weathering exposure. The thermal degradation and fire resistance of unsaturated polyester (UP) and unsaturated polyester with acrylic acid (Modar) matrices reinforced with different natural fibers (including jute, flax, and sisal at a 30 volume percentage) was compared to glass fiber composites [6]. According to the results, Modar matrix composites were more resistant to temperature than UP matrix composites. Nevertheless, the fire risk was similar between them. Among the fibers, flax fibers are the most adequate to be used. They showed the best thermal resistance due to their low lignin content, a long time to ignition, and a long period before reaching the flashover. Glass fiber composites showed more flame resistance than the bio-composites, but exhibited higher emissions of CO and CO_2_. Other disadvantages reported for synthetic fibers include a limited recycling capability that increases their carbon footprint due to further CO_2_ emissions, unsustainable production processes, the high energy involved in their manufacturing processes, and their nonbiodegradability [7,8]. Composites constituted by both synthetic and natural fibers and polymer matrix have also been reported. Hemp as continuous fiber reinforcement and palm shell and coconut shell powders as particle reinforcements were used to obtain hybrid epoxy matrix composites [9]. The researchers showed that the inclusion of an optimum amount of coconut shell and palm shell powders improved tensile strength, failure strain, flexural strength, resistance to shock, and hardness. Consequently, these composites may be used in structural applications. Thus, the addition of fibers and particles obtained from agroindustrial residues on polymer matrix composites may either increase or decrease properties. This dispersed behavior exhibited by natural fibers is associated with growing conditions and plant maturity at harvest, poor fiber extracting and processing methods, and differences in farming practices, among others. Despite this, natural fibers are low-cost, biodegradable, originate from renewable resources, exhibit low density and have other characteristics that make them attractive for lightweight applications. Particularly, weak interfacial bonding emerges as a common issue in natural fiber–polymer matrix composites, promoting variability in mechanical properties and low load transfers from matrix to fibers [7]. Some characteristics of natural fibers are considered relevant to increase interfacial bonding, such as rough and porous surfaces, high specific surface area, chemical composition (the content of hemicellulose, cellulose, and lignin), and crystallinity [1,2,5,7]. Efforts towards enhancing natural fiber–polymer matrix adhesion are mandatory to overcome these adversities.

The region of Córdoba in Colombia mainly bases its economy on agriculture. Cashew plantation is one of the crops grown in the region. Besides the nut, a rich-phenol liquid known as cashew nutshell liquid (CNSL) is extracted from cashew nutshells. This liquid provides essential components used in the fabrication of sustainable and green products, such as resins, rubbers, biodiesel, and insecticides [10,11]. Other derivative and value-added products obtained from cashew nutshell, cashew apple, and cashew biomass can be found in the literature [12,13,14,15,16,17]. Unfortunately, Córdoba’s farmers extract only the nuts, and the cashew nutshell residues are not used. Occasionally, these residues are improperly disposed of in open fields, causing environmental damage associated with the pH of the anacardic acid from the cashew nutshells [18]. Moreover, these residues are available at low or no cost because they are considered waste material, and fillers fabricated from cashew nutshells are not commercially available in the market. Therefore, a feasible alternative to convert these agroindustrial residues into valuable filaments for 3D printing arises, promoting economic and environmental benefits. Although research has been oriented toward the isolation and application of cellulose, CNSL, anacardic acid [16,19,20,21,22,23], and biochar production from cashew nutshells [24,25], few studies have been developed on evaluating their behavior in polymer matrix composites [26], and no work was found on their use on filaments for 3D printing.

Additive manufacturing techniques such as 3D and 4D printing comprise a promissory set of material processes and allow the fabrication of shape-memory materials [27], biocompatible materials [28], blending materials [29], filler powder-reinforced materials [30], and continuous or discontinuous fiber-reinforced composite materials [31], among others. Many high-value characteristics such as sustainability, assembling complexity, and flexibility are present in these manufacturing processes [32,33]. Particularly, the development of filaments containing cashew nutshell particles is not only a novel approach for the development of bio-composites but also allows the possibility of combining the biodegradability of natural fibers and PLA with the precise component fabrication achieved by 3D printing according to customer requirements [34,35]. In this work, cashew nutshell particles were prepared and their potential use as filler for filaments for 3D printing was assessed in terms of surface characteristics, apparent density, thermal stability, and content of hemicellulose, cellulose, and lignin. Furthermore, single-filament tensile tests were carried out on filaments fabricated with polylactic acid and three different mass percentages of cashew nutshell particles.

## 2. Materials and Methods

Cashew nutshells were kindly provided by Asopromarsab, a producer organization located in Chinú, Córdoba, Colombia, as shown in Figure 1. Cashew nutshells were received after manual removal of the nut. No chemical, physical, or any other treatment was carried out on them. From this point on, this original condition is referenced as the as-received condition.

Cashew nutshells were dried at 250 °C for 30 min in a muffle furnace and air-cooled to room temperature. Shore D hardness was measured according to the ASTM D2240 standard [36], and afterwards, the nutshells were submitted to a grinding process.

### 2.1. Apparent Density and Morphology Analysis

Apparent density was calculated using a predefined volume container of 123.45 cm^3^, a scale of 0.001 g, and the following equation:(1)$$\rho_{a}=\frac{m}{v}$$
where m is the mass of the particles deposited in the container (g), v is the volume of the particles including internal pores (cm^3^), and $\rho_{a}$ is the apparent density (g/cm^3^). In addition, the morphology, surface characteristics, and microstructure of the ground material were assessed by scanning electron microscopy (SEM).

### 2.2. Bromatological Analyses

Bromatological analysis was developed on the dried-cashew nutshells. Mass percentages of dry matter, ashes, acid detergent fiber (ADF), neutral detergent fiber (NDF), and cellulose were determined according to standard methods proposed by AOAC International [37]. Equations (2) and (3) were used to calculate the mass percentages of lignin and hemicellulose:(2)Lignin=ADF−Cellulose
(3)Hemicellulose=NDF−ADF

### 2.3. ATR-FTIR and TGA Analyses

Attenuated total reflectance Fourier-transform infrared (ATR-FTIR) spectra were collected using a resolution of 4 cm^−1^ from 4000 to 500 cm^−1^. Thermogravimetric analyses (TGA) were carried out from 24 °C to 900 °C using a heating rate of 5 °C/min in a nitrogen atmosphere. The first derivative of the TGA curve (the dW/dT curve) was also plotted to determine inflection points useful for in-depth interpretation of the thermal behavior of cashew nutshell particles. In both analyses, the as-received condition was used for comparison purposes.

### 2.4. Preparation of 3D Filaments

Three different filaments were fabricated from polylactic acid (PLA) pellets and cashew nutshell particles at 0.5, 1.0, and 2.0 weight percentages (wt%) using a single-screw extruder. The maximum value of 2.0 wt% of particles was selected to avoid nozzle clogging during the 3D printing process. The filaments were continuously produced at 190 °C, with an extrusion speed of 7 RPM and a nozzle diameter of 1 mm. PLA filament for comparison purposes was also fabricated. The nominal properties of the PLA pellets used in this work are displayed in Table 1.

### 2.5. Tensile Tests

Single-filament tensile tests were carried out to measure tensile strength and elongation at break percentage. The ends of a 100 mm-length single filament were glued on a 30 × 110 mm cardboard. The tests were conducted using a testing speed of 5 mm/min and a gauge length of 50 mm. The cross-sectional area of the filaments was calculated using the mean diameter determined by digital image analysis. Five tests were performed on each filament.

## 3. Results and Discussion

Figure 2 shows the cashew nutshell particles produced after the drying and grinding processes. Apparent density and Shore hardness were 0.337 g/cm^3^ and 39.67 ± 12.80 D, respectively. These low values agree with those reported for other natural fibers, and are desirable for bio-composite manufacturing due to the opportunity to produce lightweight and biodegradable components [38,39].

Figure 2 shows angular particles with a rough and porous surface and heterogeneous particle sizes ranging from ~7 to ~150 µm. Irregular and porous particles and even surfaces may enhance the filler matrix bonding in composites due to higher mechanical interlocking [7]. The influence of fillers on the properties of polymer matrix composites has been evaluated by some researchers [9]. They reported that the rigid and 3D irregular shapes of coconut shell particles provided mechanical support and were suitable for distributing stresses. The composites evidenced a higher ultimate strength of 22.9 ± 2.49 MPa; however, as the particles are nonporous, the mechanical interlocking was lower, resulting in lower failure strain values. In contrast, polymer matrix composites with palm shell particles showed extensive interlocking due to their porosity, promoting better failure strain and higher ductility. They also reported an increase in composite hardness due to the augmentation in particle percentage. Other researchers recommend enhancements on natural particle properties such as porosity and surface functionalization by physical or chemical methods. These characteristics might improve filler–matrix interaction and result in superior composite properties [1].

The characteristic plant cell wall microstructure is observable in cashew nutshell particles: an arrangement of cellulose microfibrils embedded in a hemicellulose and lignin matrix [8]. The major components in natural fibers include cellulose, hemicellulose, and lignin, while minor components include volatiles, ash, and lipid [40]. Essentially, cellulose microfibrils act as a skeletal component, hemicellulose surrounds them, and lignin provides binding and protection.

Table 2 summarizes the chemical composition of dried-cashew nutshells. Cellulose, hemicellulose, and lignin contents agree with those reported for several natural fibers [7].

Particularly, cellulose percentages in dried-cashew nutshell particles are comparable with hemp, jute, flax, and sisal fibers [6,7]. Cellulose is a strong, linear, and unbranched molecule composed of β-1,4-linked d-glucose polysaccharide units arranged into crystalline and amorphous regions. It constitutes about 9 to 80 wt% of lignocellulosic biomass [40]. Some benefits of cellulose in polymer matrix composites can be found in the literature. Storage modulus and thermal stability increased with the addition of cellulosic reinforcement in PLA composites [41]. Crystalline cellulose ensures a strong reinforcement in polymer matrix and improves tensile and flexural strength of polymer composites [42]. The strength of fibrillar cellulose combined with its economic advantages presents an opportunity to develop lighter and stronger composites [43].

Figure 3 shows the ATR-FTIR spectra for the cashew nutshells in the as-received condition and dried at 250 °C.

In both spectra, the typical wave numbers of cellulose, hemicellulose, and lignin previously reported for natural fibers are observed [2,44,45,46]. The valleys at 1374 cm^−1^ and 752 cm^−1^ evidence the hemicellulose and anacardic acid degradation, respectively, during the drying process. Wave numbers located at 3420 cm^−1^ and 1615 cm^−1^ are attributed to humidity absorption; nevertheless, the former has been assigned to crystalline cellulose [45,46]. It is possible that during the drying treatment, crystalline cellulose was formed. Many studies have been carried out in obtaining microcrystalline cellulose by alkaline or acid treatments. For instance, a conversion to crystalline cellulose from cellulose I in banana rachis treated with an 18 wt% KOH solution was reported [46]. The FTIR spectra showed two bands located at 3487 cm^−1^ and 3442 cm^−1^ associated with intramolecular hydrogen bonding in cellulose II due to changes in the hydrogen bonding system. This transition was also confirmed by X-ray diffraction and electron diffraction coupled with transmission electron microscopy. However, transformation to crystalline cellulose involving temperature treatments was not found in the literature. Further research must be done to understand this phenomenon. Table 3 shows the indexation carried out in this work.

The thermal behavior of cashew nutshells in the as-received condition and dried at 250 °C is shown in Figure 4.

The as-received condition shows four significant changes around 250 °C, 320 °C, 350 °C, and 430 °C. This behavior is consistent with the analyses described by other researchers [19,20,24,47,48], and can be associated with the following stages.

Stage I (24–200 °C): Release of the moisture absorbed between 24 °C and 120 °C [22], and decarboxylation reaction of anacardic acid around 177 °C [20,47]. This stage shows an 8 wt% reduction.

Stage II (200–375 °C): The weight reduction at this stage is attributed to three factors: 1. hemicellulose degradation from ~220 °C to ~300 °C [24,47], 2. CNSL decomposition at ~275 °C [47,48], and 3. the beginning of cellulose degradation at ~280 °C [17,22,35]. This stage exhibited a 32.4 wt% reduction.

Stage III (375–600 °C): Cellulose degradation carries on until 450 °C [19,47], and around this temperature, lignin degradation occurs [47]. From this point on, the weight percentage stabilized due to the decomposition of the material to residual char. This stage exhibited a higher weight percentage reduction of 40.66%. In contrast, on dried-cashew nutshells, changes at ~320 °C and ~430 °C only were identified, corresponding to cellulose and lignin degradation, respectively. Both moisture release and anacardic acid decarboxylation reaction occurred during the drying process at 250 °C. It is worth mentioning that ABS and PLA are the most used polymers with natural fiber fillers and their printing temperature range is from 130 °C to 250 °C [35]. These temperatures are lower than the degradation temperature of cellulose; therefore, the advantages and characteristics of dried-cashew nutshell particles discussed in this work might remain during filament manufacturing and the 3D printing process.

Figure 5 shows the PLA–cashew nutshell particle filaments fabricated in this work. Broadly speaking, the filaments exhibited an even and regular surface and were free of perceptible protuberance, voids, or cross-sectional area changes. Very few particles were observed at the surface: most of them were homogeneously dispersed throughout the filament volume. The filament diameter was 1.05 ± 0.02 mm, 1.08 ± 0.01 mm, and 1.28 ± 0.01 mm for 0.5, 1.0, and 2.0 wt% of particles, respectively.

Figure 6 shows the tensile strength as a function of elongation percentage, while Table 4 summarizes the tensile strength and elongation at break of PLA filaments and PLA–cashew nutshell particle filaments.

Cashew nutshell particle addition increased both tensile strength and elongation at break in every percentage when compared to PLA filament. The highest values of these properties were exhibited by the 2.0 wt% particle filaments. Particularly, elongation at break percentage increased ~3.36 times, evidencing a significant improvement in ductility, which might reduce the brittle response of PLA when submitted to impact conditions. Furthermore, the tensile strength showed by PLA–cashew nutshell particle filaments is greater than those reported for kenaf fiber–PLA filaments [49] and similar to crab shell–PLA filaments [50]. However, the highest elongations at break shown by those filaments were ~3.6% and ~4.2% respectively, values extremely low in comparison to the 14.1 ± 0.08% shown by 2.0 wt% PLA–cashew nutshell particle filaments fabricated in this research. Regarding other fillers in PLA matrix, wood reduces the modulus of elasticity and tensile strength of filaments by about 50%, and metallic particles including Fe, Cu, Al, and bronze have a negligible or negative impact on tensile strength, elongation at break, and flexural properties. In contrast, carbon nanotubes increase tensile strength and modulus of elasticity by around 50% and 60% [51]. However, a cost–benefit analysis should be carried out to determine whether this remarkable increase in mechanical properties is worth it.

The above characteristics for the PLA–cashew nutshell particle-reinforced composites show the feasibility of this new composite material as an alternative for 3D printing filaments. Among these characteristics are low density, surface characteristics, thermal stability, cellulose content, and homogeneous particle dispersion along the PLA matrix. In addition to these characteristics, other advantages are as follows. First, cashew nutshell is a waste material, not properly disposed of or used in many countries, and thus its use for 3D printing filaments is good for the environment and for the economy as well (as a new process can generate employment in the region and perhaps reduce costs depending on the application). Second, cashew nutshell particles significantly reinforced PLA with the particle loading used, as shown in Figure 6. This is very important for filaments, since particles can deteriorate the composite properties, particularly under tensile applications.

A complete characterization of bio-composites fabricated with these filaments, including printing ability, microstructural features, and mechanical properties, will be presented in a further work.

## 4. Conclusions

Cashew nutshell particles were produced by drying and grinding processes. Their potential use as filler for filaments for 3D printing was assessed in terms of surface characteristics, apparent density, thermal stability, and content of hemicellulose, cellulose, and lignin. Three different filaments were fabricated from polylactic acid pellets and cashew nutshell particles at 0.5, 1.0, and 2.0 wt%, and then single-filament tensile tests were carried out on them. The main conclusions can be summarized as follows.

Drying and grinding processes produced irregular, rough, and porous cashew nutshell particles with a Shore hardness of 39.67 ± 12.80D, an apparent density of 0.337 g/cm^3^, and size particles ranging from ~7 to ~150 µm. The surface characteristics of the particles are desirable because they provide a greater adhesion force due to mechanical interlocking at the interface of natural fiber and polymer matrices. The hardness and low apparent density of the particles encourage their use in the fabrication of light and biodegradable components.

The thermal stability of the cashew nutshell particles was around 320 °C; therefore, their characteristics might remain during filament production and subsequently 3D printing of polymer matrix composites.

Cashew nutshell particles exhibited the characteristic microstructure observed in natural fibers; an arrangement of cellulose microfibrils embedded in a hemicellulose and lignin matrix. The mass percentages of cellulose, hemicellulose, and lignin were 64.57, 0.70 and 24.31, respectively. Cellulose within the structure may enhance the mechanical strength and thermal stability of cashew nutshell particle–polymer matrix filaments. A possible transformation to crystalline cellulose might occur during the drying process at 250 °C. Additional research is encouraged to gain a better understanding of this phenomenon.

The filaments fabricated with polylactic acid and cashew nutshell particles are free of perceptible defects and exhibit a homogeneous distribution of particles along them. In addition, these filaments showed higher tensile strength and elongation at break in comparison to polylactic acid filament. The 2.0 wt% polylactic acid–cashew nutshell particle filaments showed the highest tensile strength and elongation at break, with 64.20 ± 2.28 MPa and 14.1 ± 0.08%.

These results suggest that the cashew nutshell particles prepared in this work are promising as natural fillers for polylactic acid matrix filaments for 3D printing.

## 5. Patents

A national utility patent application has been submitted to the Superintendency of Industry and Commerce (SIC) in Colombia: NC2023/0005743.

## Figures and Tables

**Figure 1 polymers-15-04347-f001:**
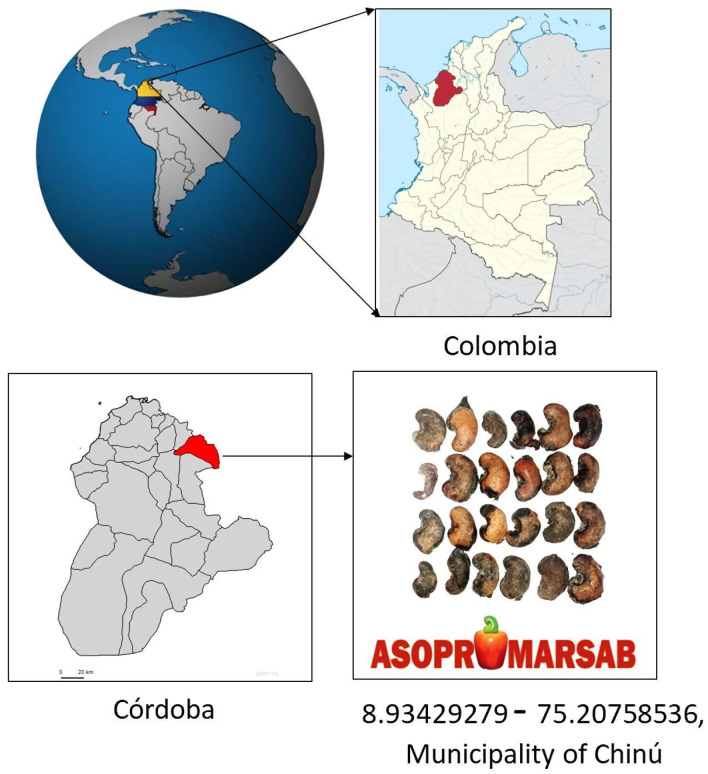
Localization of Asopromarsab and cashew nutshells used in this work.

**Figure 2 polymers-15-04347-f002:**
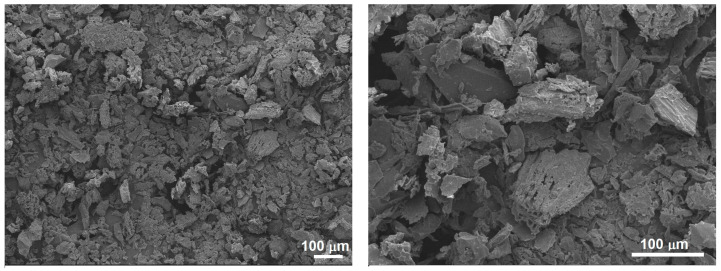
Morphology of cashew nutshell particles after drying and grinding processes. Scanning electron microscopy images.

**Figure 3 polymers-15-04347-f003:**
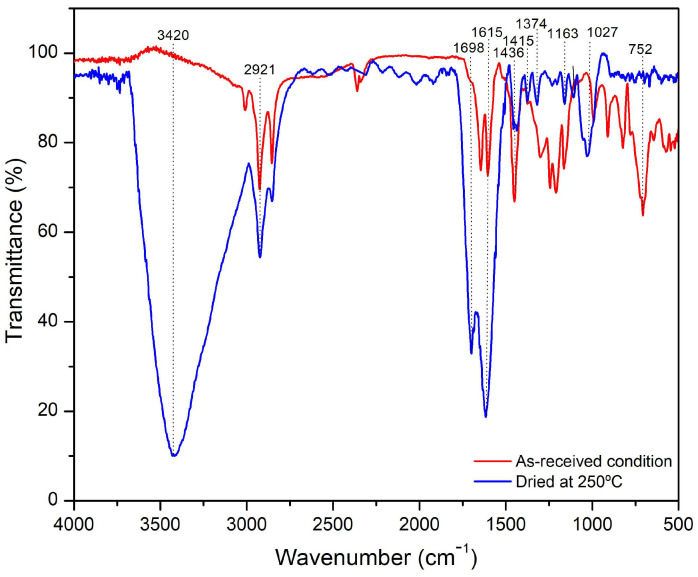
ATR-FTIR spectra for the cashew nutshells in the as-received condition and dried at 250 °C.

**Figure 4 polymers-15-04347-f004:**
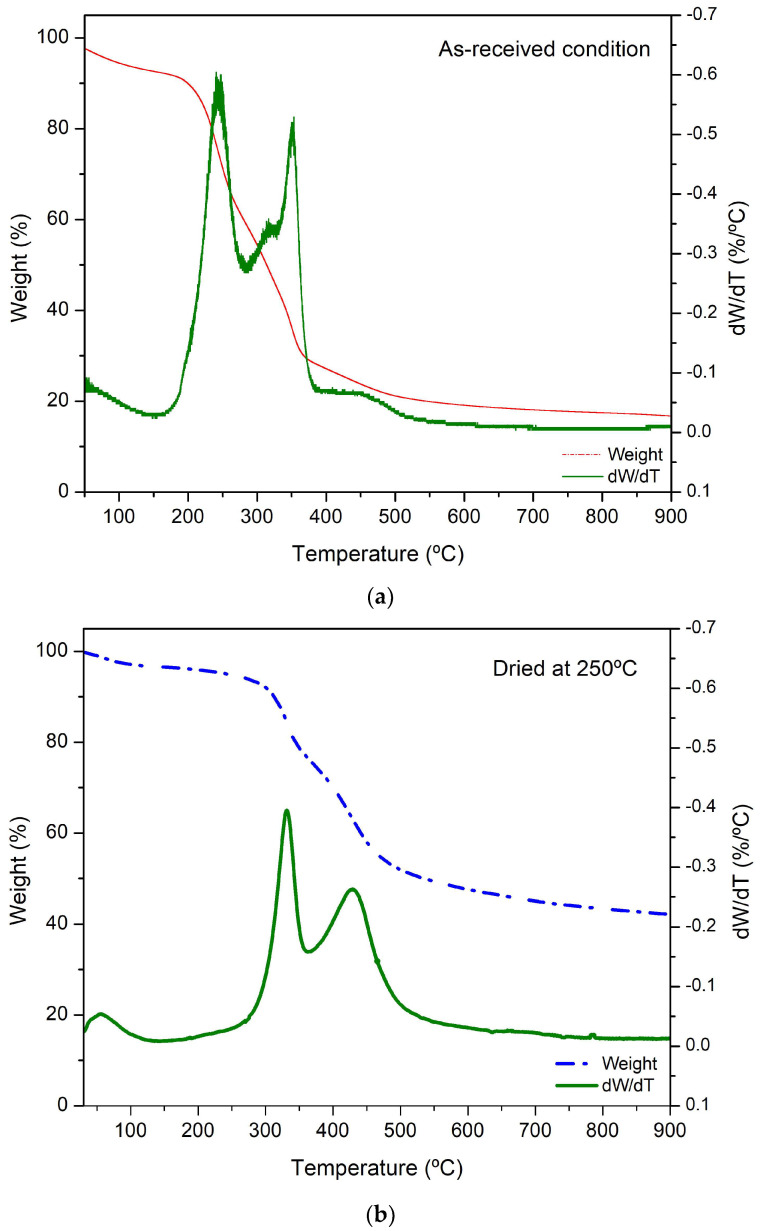
TGA and dW/dT curves of cashew nutshells in (**a**) as-received condition and (**b**) dried at 250 °C.

**Figure 5 polymers-15-04347-f005:**
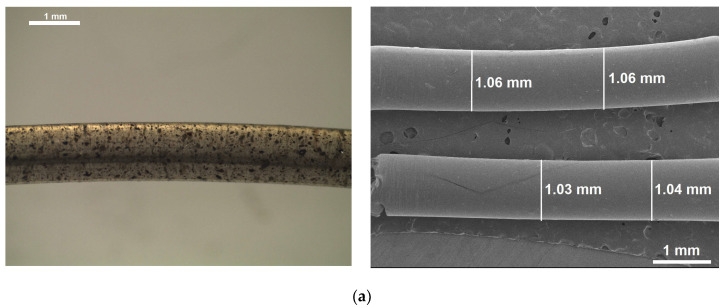
PLA–cashew nutshell particle filaments, (**a**) 0.5 wt% of particles, (**b**) 1.0 wt% of particles and (**c**) 2.0 wt% of particles. Left side: stereo-optical microscopy images. Right side: scanning electron microscopy images.

**Figure 6 polymers-15-04347-f006:**
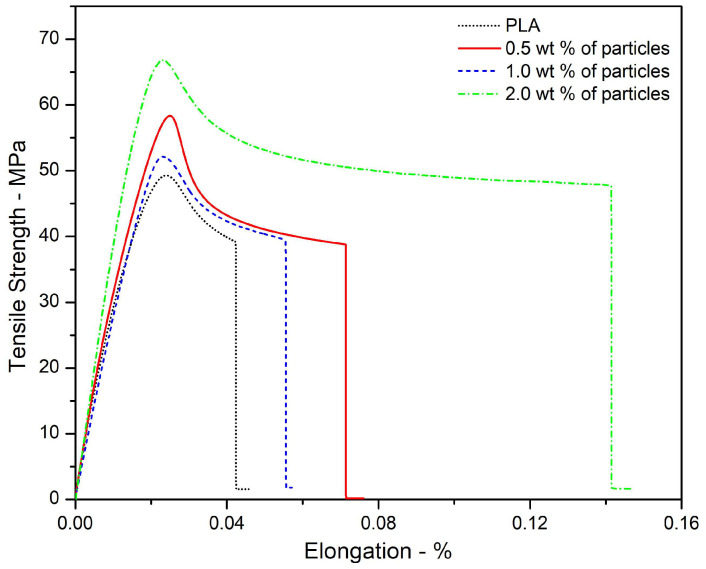
Tensile strength as a function of elongation percentage of PLA filaments and PLA–cashew nutshell particle filaments.

**Table 1 polymers-15-04347-t001:** Nominal properties of the PLA pellets used in this work.

Properties	Values
Modulus of elasticity	3450 MPa
Tensile strength	63 MPa
Tensile strain at tensile strength	4%
Tensile stress at break	44 MPa
Tensile strain at break	10%
Melting temperature	>155 °C
Density	1.25 g/cm^3^

**Table 2 polymers-15-04347-t002:** Chemical composition of cashew nutshells dried at 250°.

Component	Mass Percentage	Method
Dry matter	97.39	AOAC 930.39
Ashes	2.01	AOAC 942.05
ADF	88.88	AOAC 973.18
NDF	89.58	AOAC 2002.04
Lignin	24.31	Equation (1)
Cellulose	64.57	AOAC 973.18
Hemicellulose	0.70	Equation (2)

**Table 3 polymers-15-04347-t003:** ATR-FTIR indexation for the cashew nutshells in the as-received condition and dried at 250 °C.

Wave Number (cm^−1^)	Functional Group	Components	Observation
3420	–OH	Humidity absorption, possible cellulose ii formation	Hydrogen bonding
2921	C–H	Cellulose, hemicellulose, lignin	Aliphatic group
1698	C=O	Hemicellulose, lignin	Ester and acetyl groups of polysaccharides
1615	–(H)C=O	Cellulose, hemicellulose, lignin, humidity absorption	Carboxyl ions
1436	–CH	Lignin	Stretching on the aldehyde group
1415	–(Ar)C=C	Lignin	Stretching on aromatic fractions
1374	C–O	Hemicellulose, lignin	Acetyl group
1163	C=O	Lignin	Symmetric stretching of lignin
1027	C–O–C	Lignin, cellulose	Secondary alcohols, aliphatic ethers, cellulose monomer bonds
752	C–H	Lignin, anacardic acid	Bending vibrations on the aromatic group

**Table 4 polymers-15-04347-t004:** Tensile strength and elongation at break of PLA filaments and PLA–cashew nutshell particle filaments.

Filament	Tensile Strength MPa	Elongation at Break %
PLA	47.30 ± 6.72	4.2 ± 0.19
0.5 wt% of particles	58.28 ± 5.0	7.8 ± 0.07
1.0 wt% of particles	52.03 ± 5.17	5.6 ± 0.05
2.0 wt% of particles	64.20 ± 2.28	14.1 ± 0.08

## Data Availability

Data used in this work are confidential.

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
