# Peer review of "Cashew Nutshells: A Promising Filler for 3D Printing Filaments"

_polymers, 2023, doi:10.3390/polym15224347_

Round 1

Reviewer 1 Report

Comments and Suggestions for Authors

What is the use of cashew nutshells? Why is this filler used? Is this filler commercially available in the market? Its advantage over other metal, ceramic, carbon-based, and biocompatible fillers should be mentioned.

The language of the manuscript should be thoroughly edited.

The way of referencing in the introduction should be modified. The use of general sentences with more than four references can be seen in the first paragraph of the introduction.

The abstract is incompletely written and needs to be corrected. The steps of the article should be mentioned first, and then the results should be presented quantitatively and qualitatively. Finally, the most important achievements should be mentioned.

The number of reviewed articles is very small. The introduction is very shallow and brief. Also, the first paragraphs presented are primarily general and general information. At the end of the introduction, a suitable summary of the importance of the present issue should be provided.

Use the following resources to deepen the introduction and discussion. Shape memory performance assessment of FDM 3D printed PLA-TPU composites by Box-Behnken response surface methodology. 4D printing of PET-G via FDM including tailormade excess third shape. Toughening PVC with Biocompatible PCL Softeners for Supreme Mechanical Properties, Morphology, Shape Memory Effects, and FFF Printability.

In addition to filament preparation, the most important part of presenting a new composite material in the field of 3D printing is its printing ability and mechanical properties, which have not been done in this work. How can we talk about the ability to print complex and widely used geometries in the next step?

Some parts of the results are just reporting the results, which require corrections and deepening the analysis and discussion.

The conclusion is very brief and superficial and needs to be rewritten.

Comments on the Quality of English Language

***

Author Response

Response to Reviewer 1 Comments

Journal: Polymers ISSN 2073-4360

Manuscript ID: Polymers-2648072

Title: Cashew nutshells: a promising filler for filaments for 3D printing

1. Summary

Thank for taking the time to review this manuscript. Please find the detailed responses below and the corresponding revisions in the re-submitted file. The changes and additional information were written in red font color. Please see the attachement.

2. Questions for General Evaluation

Reviewer’s Evaluation

Response and Revisions

Does the introduction provide sufficient background and include all relevant references?

Must be improved

It was improved.

Are all the cited references relevant to the research?

Must be improved

References increased from 38 to 51.

Is the research design appropriate?

Can be improved

The research design was improved.

Are the methods adequately described?

Can be improved

The methods were improved.

Are the results clearly presented?

Can be improved

The results were also improved.

Are the conclusions supported by the results?

Can be improved

The conclusions were improved.

3. Point-by-point response to Comments and Suggestions for Authors

Comments 1: What is the use of cashew nutshells? Why is this filler used? Is this filler commercially available in the market? Its advantage over other metal, ceramic, carbon-based, and biocompatible fillers should be mentioned.

Response 1: Additional information about the use of cashew nutshells and the reasons for using it as a filler were added to the introduction. This filler is not commercially available in the market. Advantages over other fillers were also included.

Comments 2: The language of the manuscript should be thoroughly edited.

Response 2: The language was revised.

Comments 3: The way of referencing in the introduction should be modified. The use of general sentences with more than four references can be seen in the first paragraph of the introduction.

Response 3: The introduction was completely rewritten. Additional information was also included.

Comments 4: The abstract is incompletely written and needs to be corrected. The steps of the article should be mentioned first, and then the results should be presented quantitatively and qualitatively. Finally, the most important achievements should be mentioned.

Response 4: The abstract was corrected according to these comments.

Comments 5: The number of reviewed articles is very small. The introduction is very shallow and brief. Also, the first paragraphs presented are primarily general and general information. At the end of the introduction, a suitable summary of the importance of the present issue should be provided.

Response 5: The references were increased from 38 to 51. The introduction was rewritten according to these comments.

Comments 6: Use the following resources to deepen the introduction and discussion. Shape memory performance assessment of FDM3D printed PLA-TPU composites by Box-Behnken response surface methodology. 4D printing of PET-G via FDM including tailormade excess third shape. Toughening PVC with Biocompatible PCL Softeners for Supreme Mechanical Properties, Morphology, Shape Memory Effects, and FFF Printability.

Response 6: These and other references were used to enhance introduction and discussion.

Comments 7: In addition to filament preparation, the most important part of presenting a new composite material in the field of 3D printing is its printing ability and mechanical properties, which have not been done in this work. How can we talk about the ability to print complex and widely used geometries in the next step?

Response 7: We agree. This topic will be addressed in another paper as stated at the end of the results and discussion chapter.

Comments 8: Some parts of the results are just reporting the results, which require corrections and deepening the analysis and discussion.

Response 8: New reference were used to enhance the quality of the paper.

Comments 9: The conclusion is very brief and superficial and needs to be rewritten.

Response 9: The conclusion was rewritten according to the comments.

Best regards,

Luis Armando Espitia Sanjuán

Reviewer 2 Report

Comments and Suggestions for Authors

The present approach in utilizing cashew nutshells as fillers to reinforce polymers for 3D printing is interesting, especially given the potential applications and the usage of a material that is considered as waste. However, the manuscript has some serious presentation problems because it reads more like an internal laboratory report than a scientific paper. The manuscript should be significantly revised in this respect as the clarity to non-experts is very low even if the experimental approach and the tackled systems are rather standard. Several syntax and grammar errors further reduce reading clarity. It is for the benefit of the authors and the journal to make the methodology and analysis more accessible to the general readership. Unless these points are adequately addressed the work could be more suitable in a more specialized journal like for example “Sustainability”.

) According to the data on tensile strength in Fig. 6 there exists a significant change in the behavior between 1 and 2% (wt). Have the authors available more samples to cover the regime between 1 and 2%? Also, how the 2% maximum value is selected?

) Some abbreviations, even when their correspondence is common knowledge, are not properly defined when first introduced like PLA.

) Physical parameters are introduced but again they are not properly explained. For example, m and v in Eq. 1. How the volume of the ground material is calculated? Since apparent density is used how the fraction of voids is included in the calculation?

Eq. 2 and 3 appear without further explanation. What is meant by “amount” ? is it with respect to mass?

What is meant by “as-received” ? Does it correspond to the “original” state from Asopromarsab after the removal of the nut?

) Two tables are marked as “Table 1”. Table 1 should be explained better with respect to columns of percentages and methods.

) In Fig. 3 showing ATR-FTIR spectra the black lines could be changed to dashed or dotted format for better clarity.

) Fig. 4, the corresponding legend is missing in the top panel. The description for both panels is telegraphic and does not clarify on the figure content. The dW/Dt is not even explained in the main text when the results of Fig. 4 are explained. For comparison purposes it could be better to compare the weight and dW/dT trends in separate panels using the original and dried cashew shells.

) Is Fig. 6 blended with a Table? This should be avoided.

) Manuscript should be checked for numerous syntax and grammar errors which hinder its clarity and reading flow.

Line 34: Do the authors mean “Positive and/or negative impact on several properties and characteristics are reported due to the addition of fibers and particles …” ?; Line 40: “department of Cordoba” is awkward, do the authors mean “region of Cordoba” ? (again in Line 41 and in panel of Fig. 1) Also it should be “mainly bases” instead of “mainly based”; Line 49: “arise” -> “arises”; Line 52: “few work has been” -> “few works have been”; Line 53: “none work it was” -> “no work was”; Line 57: “precisely” -> “precise”; Line 58: “were assess” -> “were assessed”; Line 64: “remotion” -> “removal”;  Line 102: a format error appears with respect to the reference; Table 1: “cm3” -> “cm^3” (Table 1 could be presented with a better format: in one column the property and in the other the corresponding value); Line 133: “its porosity” -> “their porosity”; Line 137: “o chemical” -> “or chemical”; Line 145: “is surrounding” -> “surrounds”; Line 161: “proposes” -> “presents”; “strong” -> “stronger”; Line 200: “shown” -> “shows”; Line 207: “carry on” -> “carries on”; Line 217: “might remained” -> “might remain”.

Comments on the Quality of English Language

The manuscript is plagued by numerous syntax and grammar errors, I have commented on some of them in my review report.

Author Response

Response to Reviewer 2 Comments

Journal: Polymers ISSN 2073-4360

Manuscript ID: Polymers-2648072

Title: Cashew nutshells: a promising filler for filaments for 3D printing

1. Summary

Thank for taking the time to review this manuscript. Please find the detailed responses below and the corresponding revisions in the re-submitted file. The changes and additional information were written in red font color. Please see the attachment.

2. Questions for General Evaluation

Reviewer’s Evaluation

Response and Revisions

Does the introduction provide sufficient background and include all relevant references?

Can be improved

It was improved.

Are all the cited references relevant to the research?

Yes

References increased from 38 to 51.

Is the research design appropriate?

Can be improved

The research design was improved.

Are the methods adequately described?

Must be improved

The methods were improved.

Are the results clearly presented?

Must be improved

The results were also improved.

Are the conclusions supported by the results?

Must be improved

The conclusions were improved.

3. Point-by-point response to Comments and Suggestions for Authors

Comments 1: The present approach in utilizing cashew nutshells as fillers to reinforce polymers for 3D printing is interesting, especially given the potential applications and the usage of a material that is considered as waste. However, the manuscript has some serious presentation problems because it reads more like an internal laboratory report than a scientific paper. The manuscript should be significantly revised in this respect as the clarity to non-experts is very low even if the experimental approach and the tackled systems are rather standard. Several syntax and grammar errors further reduce reading clarity. It is for the benefit of the authors and the journal to make the methodology and analysis more accessible to the general readership. Unless these points are adequately addressed the work could be more suitable in a more specialized journal like for example “Sustainability”.

Response 1: The presentation of the manuscript was entirely edited. The quality of the manuscript was enhanced according to the reviewers´ comments. We believe the paper has the enough clarity for both experts and non-experts and should be published in this journal, even more in this special issue of fiber reinforced thermoplastic composites.

Comments 2: According to the data on tensile strength in Fig. 6 there exists a significant change in the behavior between 1 and 2% (wt). Have the authors available more samples to cover the regime between 1 and2%? Also, how the 2% maximum value is selected?

Response 2: Unfortunately, we do not have filaments containing cashew nutshell particles between 1 and 2 wt %. The maximum 2 wt % of particles were used to avoid nozzle clogging during 3D printing process. This information was added to materials and methods chapter.

Comments 3: Some abbreviations, even when their correspondence is common knowledge, are not properly defined when first introduced like PLA.

Response 3: All the abbreviations were properly defined at the first time of appearance.

Comments 4: Physical parameters are introduced but again they are not properly explained. For example, m and v in Eq. 1. How the volume of the ground material is calculated? Since apparent density is used how the fraction of voids is included in the calculation?

Response 4: A better explanation of equation 1 was included in materials and method chapter.

Comments 5: Eq. 2 and 3 appear without further explanation. What is meant by “amount”? is it with respect to mass?

Response 5: It is about the percentages of each component. A specific reference was included for a better understanding of these calculations.

Mary Beth Hall, David R. Mertens, Comparison of alternative neutral detergent fiber methods to the AOAC definitive method, J. Dairy Sci. 106:5364–5378, (2023). https://doi.org/10.3168/jds.2022-22847.

Comments 6: What is meant by “as-received” ? Does it correspond to the “original” state from Asopromarsab after the removal of the nut?

Response 6: Yes, it does. This information was added to materials to materials and methods chapter.

Comments 7: Two tables are marked as “Table 1”. Table 1 should be explained better with respect to columns of percentages and methods.

Response 7: This was corrected according to reviewer´s suggestion. See response to comments 5.

Comments 8: In Fig. 3 showing ATR-FTIR spectra the black lines could be changed to dashed or dotted format for better clarity.  

Response 8: The black lines were changed to dotted lines.

Comments 9: Fig. 4, the corresponding legend is missing in the top panel. The description for both panels is telegraphic and does not clarify on the figure content. The dW/Dt is not even explained in the main text when the results of Fig. 4 are explained. For comparison purposes it could be better to compare the weight and dW/dT trends in separate panels using the original and dried cashew shells.

Response 9: Figure 4 was edited for a better understanding. The explanation of the dW/dT was added to materials and methods chapter. The first derivative of TGA curve (dW/dT) is useful for a better interpretation of thermal behavior of materials. Typically, both curves are presented in the same figure, and we believe that the analysis and interpretation are easier in that way.

Comments 10: Is Fig. 6 blended with a Table? This should be avoided.

Response 10: Figure and table are presented separately.

Comments 11: Manuscript should be checked for numerous syntax and grammar errors which hinder its clarity and reading flow.

Line 34: Do the authors mean “Positive and/or negative impact on several properties and characteristics are reported due to the addition of fibers and particles …” ?; Line 40: “department of Cordoba” is awkward, do the authors mean “region of Cordoba” ?(again in Line 41 and in panel of Fig. 1) Also it should be “mainly bases” instead of “mainly based”; Line 49: “arise” -> “arises”; Line52: “few work has been” -> “few works have been”; Line 53: “none work it was” -> “no work was”; Line 57: “precisely” -> “precise”; Line58: “were assess” -> “were assessed”; Line 64: “remotion” ->“removal”;

Line 102: a format error appears with respect to the reference; Table 1: “cm3” -> “cm^3” (Table 1 could be presented with a better format: in one column the property and in the other the corresponding value); Line 133: “its porosity” -> “their porosity”; Line 137: “o chemical” -> “or chemical”; Line 145: “is surrounding” -> “surrounds”; Line 161: “proposes” -> “presents”; “strong” ->“stronger”; Line 200: “shown” -> “shows”; Line 207: “carry on” ->“carries on”; Line 217: “might remained” -> “might remain”.

Response 11: English language was revised and corrected. Colombia is dived into 32 departments, as you can see in this link https://en.wikipedia.org/wiki/Provinces_of_Colombia. However, the word department was deleted to avoid misunderstanding.

Best regards,

Luis Armando Espitia Sanjuán

Round 2

Reviewer 2 Report

Comments and Suggestions for Authors

The manuscript has been improved with respect to clarity and reading flow. However, some questions and comments have remained unanswered as the authors seem to have hurried their response and the revised version. An indication to this is line 184 where “Error! Reference source not found..” still appears, which is not acceptable.

) The authors insist on connecting the present work with 3D printing. However, such connection appears quite vague. There should be a section dedicated on describing how the observed trends of cashew nutshell filling would make the filaments as better candidates for printing application.

) The manuscript, while definitely improved with respect to the original version, is still plagued by clarity problems as some sentences are very long. Some examples: Line 85 “The scattered … applications” (what is meant by “scattered”? Could it mean “unpredictable”?)

) The questions about to which quantity the percentages refer is still unanswered in the main text (section 2.2) and in the corresponding column in Table 1.

) Keywords seem to be repeated with small variations for example: “3D Printing” vs. “3D Printing Filament”.

) The format of author names in references appearing in the main text should be changed.  

) The experimental method used to obtain the particle morphology in Figure 2 should be reported in the legend.

) Line 254: “are observed typical wavenumbers” -> “typical wavenumbers are observed”.

Comments on the Quality of English Language

The manuscript is improved but still requires polishing.

Author Response

Response to Reviewer 2 Comments – Round 2

Journal: Polymers ISSN 2073-4360

Manuscript ID: Polymers-2648072

Title: Cashew nutshells: a promising filler for filaments for 3D printing

Thank for taking the time to review this manuscript. Please find the detailed responses below and the corresponding revisions in the re-submitted file. The changes and additional information were written in red font color. Please see the attachment.

Comments 1: The manuscript has been improved with respect to clarity and reading flow. However, some questions and comments have remained unanswered as the authors seem to have hurried their response and the revised version. An indication to this is line 184 where “Error! Reference source not found..” still appears, which is not acceptable.

Response 1: Please see comments/response 4. The error sentence was deleted and will not appear again.

Comments 2: The authors insist on connecting the present work with 3D printing. However, such connection appears quite vague. There should be a section dedicated on describing how the observed trends of cashew nutshell filling would make the filaments as better candidates for printing application.

Response 2: Thank you for the clarification. The complete manuscript is about making filaments for 3D printing with a totally new composite material: PLA/Cashew nutshell particles (so, there are no previous trends of cashew nutshell), so the proposal, the aim, is a filament for 3D printing. In fact, we are in a patent process for the material and application. However, to clarify this, a paragraph regarding all these important questions was included, as follows:

“The above characteristics for the PLA/Cashew nutshell particles reinforced composites show the feasibility of this new composite material as an alternative for 3D printing filaments. Among these characteristics are low density, surface characteristics, thermal stability, cellulose content, and homogeneous particle dispersion along the PLA matrix. In addition to these characteristics, other advantages are: first, cashew nutshell is a waste material, not properly disposed of or used in many countries, thus, its use for 3D printing filaments is good for the environment, and for the economy as well (as a new process can generate employment in the region and perhaps reduce costs depending on the application). Second, cashew nutshell particles have significantly reinforced PLA with the used particle loading, as shown in Figure 6. This is very important for filament applications since particles can deteriorate the composite properties, particularly under tensile applications”.

Comments 3: The manuscript, while definitely improved with respect to the original version, is still plagued by clarity problems as some sentences are very long. Some examples: Line 85 “The scattered… applications” (what is meant by “scattered”? Could it mean “unpredictable”?).

Response 3: Many sentences were shortened. It does not mean unpredictable; it is related to the fact of being going in different directions. The word "scattered" was changed to "dispersed" for better understanding.

Comments 4: The questions about to which quantity the percentages refer is still unanswered in the main text (section 2.2) and in the corresponding column in Table 1.

Response 4: It refers to mass percentage. This was added to section 2.2 and to table 2.

Comments 5: Keywords seem to be repeated with small variations for example: 3D Printing” vs. “3D Printing Filament”.

Response 5: We agree. Keywords were modified as follows: Natural Fibers-reinforced Polymers, Cashew Nutshell Particles, 3D Printing Filament.

Comments 6: The format of author names in references appearing in the main text should be changed.

Response 6: This was corrected according to the reviewer´s suggestion.

Comments 7: The experimental method used to obtain the particle morphology in Figure 2 should be reported in the legend.

Response 7: This was corrected according to the reviewer´s suggestion, and it was also included in figure 5.

Comments 8: Line 254: “are observed typical wavenumbers” -> “typical wavenumbers are observed”.

Response 8: The sentence was rewritten as follows: “In both spectra, the typical wavenumbers of cellulose, hemicellulose, and lignin previously reported for natural fibers are observed”.

Best regards,

Luis Armando Espitia Sanjuán
